# A Deep Learning Dataloader with Shared Data Preparation

**Jian Xie**[1]    **Jingwei Xu**[1*]   **Guochang Wang**[1]    **Yuan Yao**[1]    **Zenan Li**[1]
**Chun Cao**[1]    **Hanghang Tong**[2]
[1]Nanjing University    [2]University of Illinois Urbana-Champaign
{xiejian, wgchang, lizn}@smail.nju.edu.cn
{jingweix, y.yao, caochun}@nju.edu.cn
htong@illinois.edu

## Abstract

Parallelly executing multiple training jobs on overlapped datasets is a common practice in developing deep learning models. By default, each of the parallel jobs prepares (i.e., loads and preprocesses) the data independently, causing redundant consumption of I/O and CPU. Although a centralized cache component can reduce the redundancies by reusing the data preparation work, each job's random data shuffling results in a low sampling locality causing heavy cache thrashing. Prior work tries to improve the sampling locality by enforcing all the training jobs loading the same dataset in the same order and pace. However, such a solution is only efficient under strong constraints: all jobs are trained on the same dataset with the same starting moment and training speed. In this paper, we propose a new data loading method for efficiently training parallel DNNs with much flexible constraints. Our method is still highly efficient when different training jobs use different but overlapped datasets and have different starting moments and training speeds. To achieve this, we propose a dependent sampling algorithm (DSA) and a domain-specific cache policy. Moreover, a novel tree data structure is designed to efficiently implement DSA. Based on the proposed techniques, we implemented a prototype, named JOADER, which can share data preparation work as long as the datasets are overlapped for different training jobs. We evaluate the proposed JOADER, showing a greater versatility and superiority of training speed improvement (up to 200% on ResNet18) without affecting the accuracy.

## 1   Introduction

The rapid development of deep learning frameworks and tools [2, 21] has enabled and facilitated researchers and practitioners in a multitude of disciplines (e.g., physics [27], chemistry [17], biology [11], and Earth science [22] in addition to computer science) to start developing their own DNN models solving various problems at hand. In such cases, they usually need to simultaneously run multiple training jobs on the same dataset or overlapped datasets[2], due to the purpose of model selection [14], hyper-parameter tuning [5], network architecture searching [24], etc.

Although widely used, the training of DNNs is usually time-consuming and tricky, which may affect the development efficiency, especially for general users from fields other than artificial intelligence. In literature, various approaches have been proposed to reduce the DNN training time such as data preparation optimization [3], communication overhead reduction [9, 8, 28], GPU memory optimization [7, 15, 25], and compiler-based operator optimization [6, 12, 16].

---

[*]Corresponding author.
[2]This could happen when we train models on different datasets sharing a particular portion of data, e.g., running the full experiment on ImageNet and further tuning parameters on TinyImageNet [10]).

36th Conference on Neural Information Processing Systems (NeurIPS 2022).

Different from the above existing work, our focus is on the data preparation aspect, or more specifically, the data loading aspect, in the context of parallel DNN training. On the one hand, Mohan et al. [18] have observed that data preparation, which consists of data loading and the subsequent data preprocessing, occupies a significant portion of parallel DNN training time. This is due to the fact that multiple training jobs on the same data copy essentially create redundant data preparation efforts. Ideally, to reduce the load of hardware and boost the training efficiency, the prepared data should be cached in memory and serve for multiple jobs. However, the cache may thrash with a high probability as each job adopts a fully random shuffling strategy that independently shuffles the dataset in each epoch. On the other hand, it is until recently have some studies investigated the shuffling process itself [19, 18, 20]. These studies investigate to what extent the fully random shuffling is required and have observed that various shuffling strategies breaking the full randomness still yield competitive results. We name such randomness as *correlated randomness* as opposed to the full randomness. For example, prior work [18, 19] proposed to solve the cache thrashing issue by enforcing that all jobs iterate over the dataset in the same order and pace, and observed no accuracy loss. However, such a solution maximizes its performance under some strong constraints, i.e., all training jobs have to start simultaneously, use the same dataset, and have similar training speeds.

This paper proposes a new data loading method for training multiple parallel jobs. Specifically, we aim to relax the constraints of prior work in three aspects: 1) all jobs can start and end freely, 2) different datasets can be used as long as they are partially overlapped, and 3) jobs' training speeds may vary widely. For the above purposes, we first propose the *dependent sampling algorithm* (DSA) to schedule the sampling for each job while ensuring their correlated randomness. DSA is inspired by the correlated sampling theory [4] and it mainly involves three steps: 1) divide the dataset into two parts of intersection set and difference set, 2) correlate the selections between intersection and difference, and 3) share some operations for the intersection set to improve locality. Furthermore, we design a domain-specific cache policy *RefCnt* that is tailored for parallel DNN training. It is aware of the fact that all jobs iterate over the dataset once in an epoch, and thus it can evict data that is least likely to be used in the near future. Finally, we design a novel data structure *dependent sampling tree* to efficiently implement the DSA by elaborately organizing the intersection sets and difference sets. Based on the above techniques, we implement a prototype system named JOADER and integrate it into PyTorch for evaluation. The evaluation results show that JOADER can boost training efficiency in more flexible cases. For example, it achieves up to 200% training speedup when models of different sizes are parallelly trained without affecting the accuracy.

The main contributions of this paper inlcude:

- A new dataloader for parallel DNN training on overlapped datasets.
- A sampling algorithm to increase the sampling locality with guaranteed randomness. The algorithm reaches the global optima when there are two training jobs.
- A domain-specific cache policy and a novel tree-based data structure for efficient implementation.
- Experimental evaluations demonstrating the performance improvements.

The rest of this paper is organized as follows. Section 2 introduces the background information. Section 3 describes the dependent sampling algorithm and our cache policy, followed by the description of the dependent sampling tree in Section 4. Section 5 presents the evaluation of the implemented dataloader JOADER. The limiations are discussed in Section 6 and the paper is concluded in Section 7.

## 2 Background

### 2.1 Problem Statement

In each epoch, a DNN training job sends data requests to the storage device and passes through the dataset in random order. To simply the presentation, we assume that each data request involves only one data point/element, and we name the data access order over all the data elements as the *access path* for each epoch. Consider an example in Figure 1, where we assume two training jobs are running, and the two sampled data elements ('e1' and 'e2') requested by the two jobs are packaged and prepared together in one *round*.

Cache can hold the prepared data to serve the future duplicated requests. To fully utilize the cache when multiple DNN models are being trained simultaneously in a server, one could generate a

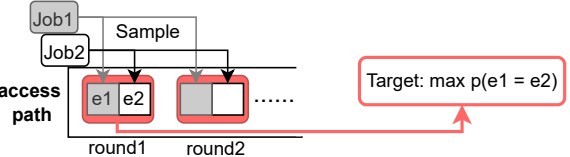

Figure 1: The access path of two training jobs. The target is to maximize the probability of sampling the same element in each round for all the jobs.

cache-friendly access path by encouraging all jobs to request the same data within a short period, i.e., improving the sampling locality in each round. Meanwhile, we still need to ensure the correlated randomness for each job. To this end, we formulate the problem as follows:

**Problem 1** *Assume there are $n$ training jobs on overlapped datasets, where the $i$-th job $J_i$ trains a model on dataset $D_i$ with cardinality $|D_i|$. In each round, $p(e_j^i) = \frac{1}{|D_i|}$ indicates the probability of data element $e_j$ being picked via sampling uniformly at random by $J_i$. For a given round, the goal is to maximize $p(e^1 = e^2 = ... = e^n)$, while maintaining $p(e^i) = \frac{1}{|D_i|}$.*

## 2.2 Related Work

**Dataloader for multiple training jobs.** Cerebro [19] is proposed to avoid redundant data preparation work for the task of model selection. It partitions the dataset across the servers into clusters and hopes the models iterate over data from one server to another instead of shuffling data. Although it can help avoid redundant data preparation work across servers, the work is still repeated on a single server. CoorDL [18] provides a more general solution for hyper-parameter optimization. In one turn, it distributes one batch to each job and caches the batch into the buffer for future usage. The batches are evicted until all jobs have consumed them in an epoch. However, CoorDL still has some limitations: 1) all jobs must be set up with the same batch size; 2) when the jobs have different training speeds, the faster jobs must wait for the slower ones; 3) CoorDL cannot handle situations where the jobs arrive at different moments or are trained on different (but overlapped) datasets.

**Domain-specific caching.** Crafting a cache for a specific domain is not a new idea [23]. For example, Quiver [13] uses local solid-state drive (SSD) caches to eliminate the impact of slow reads from the remote storage. However, it is too expensive to prepare enough memory against cache thrashing for DNN training, especially considering the fact we need to cache not only the previous reading but also the results of data preprocessing. In this paper, we propose a well-designed sampling algorithm to improve the sampling locality with a small cache.

**Correlated sampling.** As a theoretical problem, correlated sampling [4] aims to minimize the probability of two sampling results being equal, given that the two players are sampling uniformly at random from two subsets of the same set. Some sampling strategies [26] give the optimal solution for two players. In this paper, we practically solve a related problem by designing an algorithm and a novel data structure for efficient implementation while enjoying their theoretical results.

## 3 Dependent Sampling Algorithm

### 3.1 Algorithm Design

To solve Problem 1, we propose a dependent sampling algorithm (DSA) to maximize the sampling locality while ensuring correlated randomness. In the following, we first present the algorithm in the two-job case and then extend it to the n-job case.

**Two-job Case.** The proposed *dependent sampling algorithm* (DSA) involves three steps: 1) intersection calculating, 2) dependent selecting, and 3) shared sampling. For better illustration, we assume two jobs $J_1$ and $J_2$ are being trained on two overlapped datasets $D_1$ (the yellow circle) and $D_2$ (the green circle), respectively, as shown in Figure 2.

First, we divide the datasets into three subsets: *intersection* set $I = D_1 \cap D_2$ and two *difference* sets $D_{d1} = D_1 \setminus I, D_{d2} = D_2 \setminus I$. For the default *independent sampling algorithm* (ISA), which

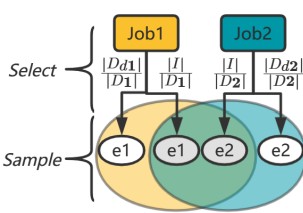 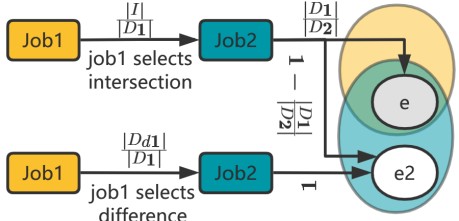 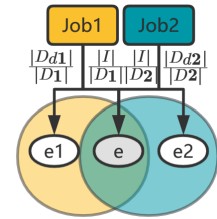

(a) *Intersection calculating*: 1) calculate the intersection and difference; 2) let job select subset first and then sample in this subset randomly.

(b) *Dependent selecting*: Job2 makes decisions conditioned on the decisions of Job1, under the constraints that the probabilities of Job2 selecting difference set and intersection are uniformly distributed according to the total probability law.

(c) *Shared sampling*: if two jobs both select the intersection, then they share the sampling results.

Figure 2: Dependent sampling algorithm in the two-job case.

is currently used by PyTorch, it processes the data sampling uniformly at random in the dataset for each job independently. As shown in Figure 2a, the ISA algorithm can be divided into two steps: *selecting* and *sampling* procedures. In the selecting procedure, each job selects between its difference and intersection sets for sampling. In the sampling procedure, each job randomly picks an element $e$ from the subset chosen in the previous step. This two-step process maintains the uniform distribution of sampling for each job.

The proposed DSA takes a careful modification in the sampling procedure. As shown in Figure 2c, if the two jobs both select the intersection set, one shares the sampling result to another. We call this modification *shared sampling*. Based on shared sampling, the probability that two jobs selecting the same element is equal to that both jobs selecting the intersection set.

To further improve locality, we need to improve the probability of both two jobs selecting the intersection, which can be achieved according to the conditional probability, as shown in Figure 2b. The independent events in the selecting procedure can be transformed into the dependent ones, making the probability of the second job conditional. That is, $J_2$ makes decisions conditioned on the decision of $J_1$.

To maximize the sampling locality, we formulate the conditional probabilities of dependent selecting in case of $|D_1| < |D_2|$. For $J_1$, the dependent selecting procedure is the same as that in ISA: selecting $I$ with the probability of $\frac{|I|}{|D_1|}$, and $D_{d1}$ with the probability of $\frac{|D_{d1}|}{|D_1|}$. The selecting procedure of $J_2$ makes the decision according to $J_1$. If $J_1$ selects $D_{d1}$, then $J_2$ must select another difference set $D_{d2}$. If $J_1$ select $I$, then $J_2$ selects $I$ with the probability of $\frac{|D_1|}{|D_2|}$ and $D_{d2}$ with probability $\frac{|D_2|-|D_1|}{|D_2|}$. Figure 2b shows the above selecting procedure. For the case of $|D_2| < |D_1|$, we just need to exchange the order of $J_1$ and $J_2$ to apply the above rules.

According to above conditional probabilities, the probability of $p(e_j^1 = e_i^2)$ is $\frac{|I|}{\max(|D_2|,|D_1|)}$, which is also the theoretical upper bound as discussed in Theorem 1.

**Theorem 1** *Assume there are two jobs $J_1$, $J_2$ sampling uniformly at random from two datasets $D_1$, $D_2$ respectively, and the elements they sampled are $e_j^1$, $e_i^2$. The probability of these two elements being equal cannot be larger than $\frac{|I|}{\max(|D_2|,|D_1|)}$ where $I = D_1 \cap D_2$.*

**N-job Case.** DSA of $n$-jobs is similar to two-jobs at the beginning. The difference happens when some jobs select difference sets. We need to go through the above procedure recursively for these jobs rather than sampling in the current difference set directly, as shown in Figure 3. The reason for recursive execution is that some data may only be shared by part of the jobs. In the extreme case, if all datasets are mutually disjoint, then the intersection in the first recursion is empty and the DSA is useless. Therefore, we need to further check whether there are shared data for part of the jobs.

The conditional probability for n-jobs that are sorted by the cardinalities of their datasets is formulated as follows. 1) The first job $J_1$ selects the intersection with the probability of $\frac{|I|}{|D_1|}$ while selecting the

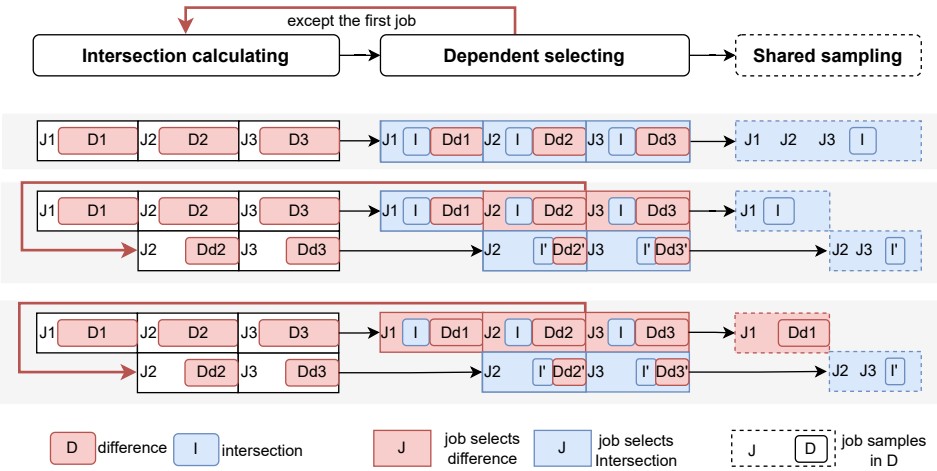

Figure 3: An illustration for the $n$-job case of DSA. We assume three jobs $\{J_1, J_2, J_3\}$ with datasets $\{D_1, D_2, D_3\}$, and there are three possible sampling cases: 1) all jobs select intersection, 2) some jobs select intersection and some select difference, and 3) all jobs select difference. Note that if there is one job selecting difference, all following jobs must select difference.

difference with the probability of $\frac{|D_{d1}|}{|D_1|}$. 2) If the job $J_{i-1}$ selects the difference set, then job $J_i$ must select the difference set. If job $J_{i-1}$ selects the intersection set, then job $J_i$ selects the intersection with the probability of $\frac{|D_{i-1}|}{|D_i|}$ while selecting the difference set with the probability of $\frac{|D_i|-|D_{i-1}|}{|D_i|}$. For the above algorithm, we can observe that if one job selects the difference, then all subsequent jobs will choose the difference, which is an important feature for the data structure in Section 4.

**Theorem 2** *Assume there are multiples jobs with the $i$-th job $J_i$ trained on dataset $D_i$. The probability of any element $e$ being picked for $J_i$ via DSA is $p(e) = \frac{1}{|D_i|}$.*

### 3.2 RefCnt Cache Policy

If we allow jobs to vary greatly in speed, then some data may be evicted before they are consumed by all jobs. Therefore, we need a cache policy that can take out the data that is least likely to be used (i.e., the data that least jobs will request) in the near future. Classical cache policies (e.g., LRU and LFU) are not suitable for deep learning scenarios, where each training job reads each data only once in an epoch. So the total number of request to each data equals to the number of jobs using the data for training. To this end, if we could record the count of data request, the number of request to this data in near future could be derived. The fewer jobs that access the data, the higher the eviction priority of that data.

Based on the above understanding, we design the caching policy *RefCnt*. We maintain a reference count for each data element. If a job will request the element in the future, we increase its reference count; if a job reads the data, we decrease its reference count. When the cache is full, the data with the lowest reference count will be evicted.

## 4 Dependent Sampling Tree

The time complexity of DSA is $O(nm)$, where $n$ is the number of jobs and $m$ is the maximum cardinality of the datasets. The worst case happens if all jobs select the difference set in each recursion, which means we need $n$ recursions for $n$ jobs. Each recursion has the time complexity of $O(m)$.[3]

---

[3] 1) The intersection calculation needs to traverse the dataset so its complexity is linear to the cardinality of the dataset. 2) In the worst case, only the first job needs to make the decision, and others follow it. Therefore, dependent selecting needs constant time. 3) Sampling only needs to pick a random number between 0 and the size of the dataset, which needs constant time.

To reduce the time complexity of sampling, we can maintain the intersections for DSA to avoid the intersection calculation in each recursion, which can make each recursion cost constant time. Consequently, we can reduce the worst-case time complexity from $O(nm)$ to $O(n)$.

We propose the dependent sampling tree for organizing the intersections. Although the number of intersections is $2^n - 1$ for $n$ sets, only $n - 1$ intersections are valuable. In DSA, we only need to calculate the intersection of datasets for those jobs that select the difference, as shown in Figure 3. When the jobs are sorted by the cardinality of their datasets, the job selecting the difference will make all following jobs select the difference sets. Therefore, the possible combinations for jobs that select the difference must be the suffixes of the sorted jobs array. For example, if there are three sorted jobs $[J_1, J_2, J_3]$, the groups of jobs choosing the difference set can only be one of the following three sets, including $[J_1, J_2, J_3]$, $[J_2, J_3]$, and $[J_3]$. Thus, the sets to be calculated are $\{D_1 \cap D_2 \cap D_3, D_2 \cap D_3\}$.

## 4.1 Tree Definition

For $n$ datasets $\{D_1, D_2, ..., D_n\}$ in ascending order with $|D_1| < |D_2| < ... < |D_n|$, we need to calculate $n - 1$ intersections $\{D_1 \cap ... \cap D_n, D_2 \cap ... \cap D_n, ..., D_{n-1} \cap D_n\}$. We build a dependent sampling tree, where each intersection is an internal vertex, and each difference set is a leaf. Meanwhile, all vertices in a path from the root to a leaf represent a complete dataset. The definition of dependent sampling tree is as follows.

**Definition 1** *Dependent sampling tree is a binary tree that stores a collection of datasets $\{D_1, D_2, ..., D_n\}$, with $|D_1| < |D_2| < ... < |D_n|$. In the tree, the root is the intersection of all datasets $I = D_1 \cap D_2... \cap D_n$. The right child is the difference of the smallest dataset $D_1 \setminus I$. The left child is a new dependent sampling tree of $\{D_2 \setminus I, D_3 \setminus I, ..., D_n \setminus I\}$. When there is only one dataset, the tree has one vertex that contains the dataset.*

If there are $n$ datasets, the height of the dependent sampling tree is $n - 1$. Figure 4a shows an example with three sets $\{D_1, D_2, D_3\}$ with $|D_1| < |D_2| < |D_3|$. In this tree, vertex A contains $D_A = D_1 \cap D_2 \cap D_3$, and vertex B contains $D_B = (D_2 \setminus D_A) \cap (D_3 \setminus D_A)$. For the leaf vertices $C$, $E$, and $F$, they maintain the other differences.

## 4.2 Tree Operations

When executing the DSA algorithm, dependent sampling tree should maintain the datasets into intersection and difference sets dynamically. Furthermore, dependent sampling tree should also support the situations when a job finishes its execution or a new job starts. In this part, we discuss the operations provided by dependent sampling tree, which are *sampling*, *deletion*, and *insertion*.

**Sampling.** The intersections of the flow in Figure 3 can be represented by one or multiple vertexes of the rightest path in the dependent sampling tree. Thus, the process of DSA is to traverse the path from the root to the most-right leaf in the tree. When traversing to an internal vertex, the jobs will decide whether to select this vertex according the conditional probability. For those jobs that do not select this vertex, they will make decisions in the right child tree.

Sometimes the intersection can be composed of multiple vertexes. For example, if only the job of $D_3$ selects difference from $A$ in Figure 4a, then it can sample from both $B$ and $F$. In these cases, the algorithm needs to first select a vertex that belongs to the intersection and then sample.

**Dataset Deletion.** When a job finishes training or is killed, we should delete its dataset from the tree. The *deletion* involves two steps: 1) deleting the corresponding leaf vertex, and 2) merging its parent vertex with its sibling vertex. For example, if we want to delete $D_2$ in Figure 4a, we need to delete the leaf vertex $E$ and merge vertex $B$ with $F$, as shown in Figure 4b. The time complexity is $O(1)$ since deleting and merging both cost constant time.

**Dataset Insertion.** The *insertion* operation depends on the cardinality of the new dataset $D$. When $D$ is smaller than the smallest set in the sampling tree, the procedure involves three steps: 1) calculating the intersection of it and the root of the tree, 2) setting the new intersection as the new root of the dependent sampling tree, and 3) setting the difference set of the old root as the right child and the difference of $D$ as the left child for the new root. For example, Figure 4c shows the new tree after

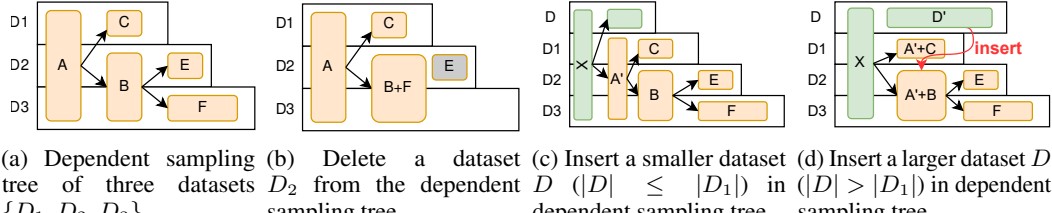

(a) Dependent sampling tree of three datasets $\{D_1, D_2, D_3\}$

(b) Delete a dataset $D_2$ from the dependent sampling tree

(c) Insert a smaller dataset $D$ ($|D| \leq |D_1|$) in dependent sampling tree

(d) Insert a larger dataset $D$ ($|D| > |D_1|$) in dependent sampling tree

Figure 4: Illustration for the proposed dependent sampling tree.

inserting a smaller dataset $D$. The new root $X$ is the intersection of $D$ and the set in vertex $A$. The old tree becomes the right sub-tree, and the difference of the dataset $D$ becomes the left leaf.

When the new dataset $D$ is larger than the smallest dataset in the sampling tree, we need to insert it recursively: 1) calculating the intersection of $D$ and root $A$ and set it as new root $X$, 2) adding the difference of $A$ to its children, 3) deleting the old root $A$ and adding their children to the new root, 4) recursively executing step 1 on $X$'s right child with the difference set of $D$ until the difference set is the smallest in the current sub-tree. An example of this procedure is shown in Figure 4d. In the worst case, the time complexity of *insertion* is $O(n|D|)$ if the new dataset $D$ is the largest [4].

## 5   Evaluation

In this section, we evaluate JOADER on ImageNet with the family of ResNet models. We denote the default dataloader strategy in PyTorch as the 'Baseline' method, and further compare JOADER with the state-of-the-art method CoorDL [18]. First, we profile the server when there are multiple training jobs to show the bottlenecks and compare JOADER with CoorDL and Baseline in the synchronous case. Second, we evaluate JOADER in asynchronous cases: 1) multiple models trained at different speeds, 2) multiple jobs arrived at different moments, and 3) multiple datasets only partially overlapped. For the other results in experiments, we summarize them as follows: 1) ablation studies show that both DSA and RefCnt improve the training efficiency , 2) the I/O speed is not the bottleneck for data loading , and 3) we train ResNet18 with PyTorch and JOADER separately to show JOADER does not affect the accuracy.

The evaluated models are the basic models with their default settings in torchvision [1], and trained on top of the PyTorch 1.6.0 DL framework. The experiments were conducted on a GPU server with two Intel Xeon Gold 5118 CPUs @ 2.30GHz (24 physical cores and 48 threads), 500GB RAM, and 6 TITAN RTX GPUs. The server ran Ubuntu 18.04 with GNU/Linux kernel 4.15.0. The disk is Symbios Logic MegaRAID SAS-3 3316 of 1GB/s read speed.

### 5.1   Performance in Synchronous Cases

In this experiment, we start training multiple ResNet18 models at the same time but with different hyper-parameters. We start from training 1 model to training 6 models on ImageNet, and profile the server for different dataloaders. The experiments show that the CPU utilization tends to be the bottleneck, as shown in Figure 5.

The CoorDL and JOADER both can reduce the CPU utilization greatly by sharing the data preparation. Additionally, it shows that JOADER can achieve close performance to CoorDL when all training jobs start at the same time and have a similar speed. Note that CoorDL is designed for such synchronous cases while JOADER can also handle asynchronous cases as we will later show. The time cost of each epoch is shown in Table 1. With 6 training jobs, JOADER saves $44.8\%$ training time with over 40% less CPU utilization, compared the default Baseline in PyTorch.

---

[4]In practice, the dataset is a bitmap set of data indices in the dataset, e.g., a unique integer id that identifies data. Therefore, the insert operation is very fast, which only takes little time compared with training time.

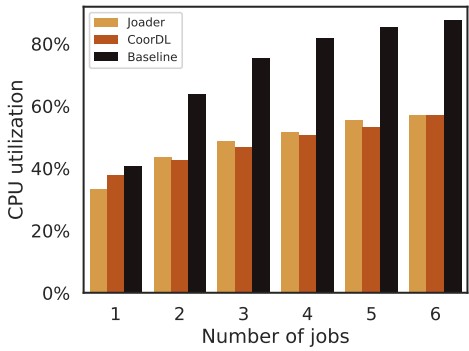

Figure 5: The CPU utilization when training multiple ResNet18 models simultaneously. CPU quickly becomes the bottleneck for the Baseline dataloader of PyTorch, while both CoorDL and JOADER can greatly reduce CPU utilization.

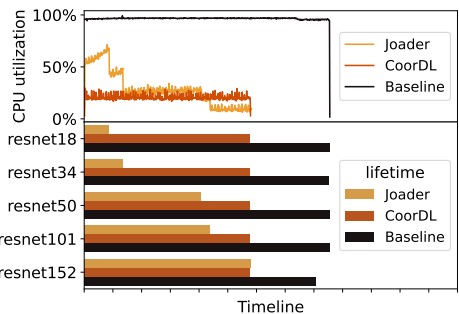

Figure 6: The CPU utilization and lifetime when training 5 models with different speeds. JOADER can reduce the CPU utilization and keep each job's own speed. In contrast, fast job must wait for the slow job for CoorDL (e.g., ResNet18 costs the same time with ResNet152).

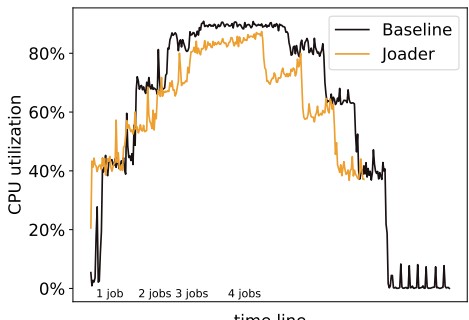

(a) The CPU utilization and lifetime when four jobs start at different moments. JOADER is faster and causes less CPU load because it can reuse the data preparation work from previous jobs.

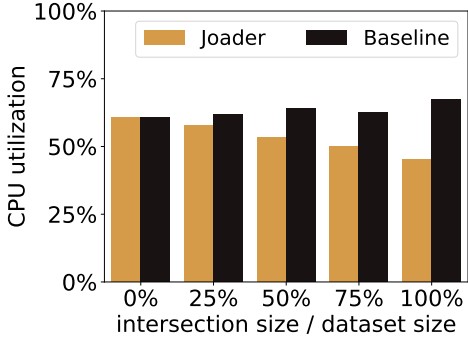

(b) The CPU utilization when datasets are partially overlapped. JOADER can reduce 10% CPU utilization when half of datasets are in the intersection set for 2 jobs.

Figure 7: Training jobs arrive at different moments and have different datasets

Table 1: The time cost of different dataloaders when training multiple ResNet18 models in an epoch.

| Number of jobs | 1 | 2 | 3 | 4 | 5 | 6 |
|---|---|---|---|---|---|---|
| CoorDL (min) | 32.82 | 31.73 | 32.73 | 34.76 | 35.28 | 37.98 |
| JOADER (min) | 32.70 | 32.49 | 33.72 | 36.32 | 37.44 | 37.55 |
| Baseline (min) | 31.74 | 35.01 | 40.04 | 52.11 | 57.67 | 67.85 |

## 5.2 Performance in Asynchronous Cases

Next, we consider three asynchronous cases when multiple models differ in training speeds, arriving moments, and datasets.

**Multiple models with different training speeds.** We first simultaneously train five models (ResNet18, ResNet34, ResNet50, ResNet101, and ResNet152) with different training speeds, and show the lifetime and CPU utilization results in Figure 6. With the Baseline dataloader provided by PyTorch, each job preprocesses the data individually so that the CPU load is extremely heavy. CoorDL enforces multiple jobs requesting the data in lockstep to share the data preparation work, and it achieves less CPU load and higher training speed than Baseline. However, the faster jobs (smaller models) must wait for the slower jobs, making CPU idle during the training period. JOADER can both reduce the CPU load by sharing the preprocessing work and keep these jobs' own training speeds.

For example, ResNet18 runs $4\times$ faster using JOADER compared with CoorDL. Furthermore, this experiment suggests that our system has less profitable for larger models because GPU tends to be the bottleneck for those models.

**Multiple models arriving at different moments.** Next, we consider the situation that multiple DNN training jobs start at different moments. Since CoorDL must wait for all the jobs to be ready for training, we only compare JOADER with Baseline in this experiment. Figure 7a shows the CPU utilization when four ResNet18 training jobs arrive at different moments. JOADER takes up less CPU load than the Baseline. This is due to the fact Baseline repeats the data preparation work for new jobs, whereas JOADER reuses the prepared data from previous jobs. As a result, JOADER saves 17.1% training time and 8.7% CPU utilization.

**Multiple models with partially overlapping datasets.** Finally, we show the CPU utilization when the two datasets only partially overlap in Figure 7b. The datasets are re-sampled from ImageNet. We use the ratio between intersection size and dataset size to measure the overlapping degree. The larger the intersection, the more data to share. We still compare with the Baseline method only since CoorDL requires all jobs having the same access path over datasets. From the figure, we can observe that the CPU utilization of the same dataset is only 60% of that of different datasets. Additionally, the CPU utilization always decreases as the intersection size increases, showing that JOADER is valid with some overlapping elements. The reason is that the DSA algorithm can get the same sampling results with the max probability and the cache policy evicts the useless data with higher priority.

# 6   Discussion

**Limitations**. Although JOADER proved its effectiveness in classic scenarios, it inevitably works under certain constraints. For the DSA, superior locality as it brings, it also breaks the strict independence across workloads. As a result, JOADER will not achieve an as good result for correlated workloads (e.g., parallel tasks aimed for variance reduction by ensembles).

Additionally, although JOADER can be applied to all machine learning tasks that need to iteratively traverse the dataset in random order, the benefit mainly depends on the cost of data reading and processing. That is, larger benefits can be obtained for tasks in computer vision, since image decoding usually takes much calculation. In contrast, smaller benifits can be obtained for NLP tasks where each data sample only contains several bytes.

**Efficiency of dependent sampling tree**.   For efficiency of the dependent sampling tree implementation, the bottleneck may come from CPU or disk I/O. Based on our experiment, disk I/O cost during training process would be masked by CPU cost if the disk I/O speed is over 350MB/s. Otherwise, the bottleneck would come from I/O. We tested our implementation under an SSD with disk I/O read speed at 1GB/s, which is a pretty common specification, and any disk with better read speed should achieve better results.

Dependent sampling tree also brings more operations on datasets, such as calculating intersection. To reduce the operations, the bitmap index is used to represent the dataset. To evaluate the time cost of dataset operations, we have conducted an additional experiment by randomly inserting 128 datasets into our system. The size of each dataset is between 1,000,000 and 2,000,000 elements (note that ImageNet contains 1,400,000 images). The average cost of inserting each dataset is 0.57 seconds.

# 7   Conclusion

Accelerating DNN training is a fundamental challenge in deep learning. This paper presents an approach to boosting training efficiency by caching data preparation work and reusing them to parallel training jobs. To prevent cache thrashing, we propose DSA to improve sampling locality with ensured randomness. We also propose a novel tree-based data structure to efficiently implement the proposed algorithm. Furthermore, a domain-specific cache policy is proposed for evicting data that is least used in the near future. Experimental results of the proposed approach show significant improvements especially in asynchronous cases compared with the existing methods.

## Acknowledgments and Disclosure of Funding

We thank the anonymous reviewers for their helpful comments. This work was supported in part by the Key-Area Research & Development Program of Guangdong Province (No. 2020B010164003), the National Natural Science Foundation of China (No. 62025202, 62172199), and the Collaborative Innovation Center of Novel Software Technology and Industrialization. Hanghang Tong is partially supported by NSF (1947135).

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
