# Appendix of A Deep Learning Dataloader with Shared Data Preparation

Jian Xie[1]    Jingwei Xu[1]*   Guochang Wang[1]    Yuan Yao[1]    Zenan Li[1]
Chun Cao[1]    Hanghang Tong[2]
[1]Nanjing University    [2]University of Illinois Urbana-Champaign
{xiejian, wgchang, lizn}@smail.nju.edu.cn
{jingweix, y.yao, caochun}@nju.edu.cn
htong@illinois.edu

## A    More Evaluations

In this section, we first present the I/O speed in the synchronous cases and asynchronous cases to show JOADER can reduce the redundant I/O work. Then, we evaluate the algorithm DSA and RefCnt separately for the ablation test. Finally, we show the loss and accuracy trace of ResNet18 in 40 epochs to demonstrate JOADER does not affect convergence speed. We denote the default dataloader strategy in PyTorch as the 'Baseline' method, and further compare JOADER with the state-of-the-art method CoorDL.

### A.1    I/O speed

In this part, we show the I/O speed in the synchronous and asynchronous cases. Note that I/O is not the bottleneck in our server (Maximum I/O speed of the server is about 1GB/s). Figure 1 shows the I/O from training one job to training six jobs. The I/O speed of Baseline tends to increase while that of CoorDL and JOADER tends to decrease, because CoorDL and JOADER can reuse the data in memory. When we train 5 jobs (ResNet18, ResNet34, ResNet50, ResNet101, ResNet152) with different speed, the I/O speed is shown in Figure 2. Although the I/O speed of CoorDL is far less than JOADER, the fast job must wait for a slow job causing inefficiency. JOADER can make jobs run at their own speeds. Figure 3a show the I/O speed for four jobs that start at different moments. Due to the redundant I/O work, the baseline has a more considerable I/O speed. For JOADER, the more data is shared, the smaller the I/O speed, as shown in Figure 3b.

### A.2    Algorithm evaluation

In this section, we first compare ISA, the default sampling algorithm in PyTorch, with DSA to show the performance in two cases: 1) datasets overlap partially, and 2) datasets vary in size. Then we further compare the RefCnt with the generic cache policy in the above cases. We evaluate these algorithms in different cache sizes and the evaluation metric is the count of *cache misses*, indicating the number of elements not read from the cache. For $n$ datasets $D_1, D_2, ..., D_n$, the maximum count of cache misses is $|D_1| + |D_2| + ... + |D_n|$ that all the data is read from the storage, while the minimum is $|D_1 \cup D_2 \cup ... \cup D_n|$ that all the repeated data is hit in cache.

We construct the following cases for evaluation[2] in Table 1:

**Dependent sampling algorithm.** In this part, we show DSA can significantly reduce the cache misses in various of scenes. To simulate various scenes, we generate multiple datasets with different

---

*Corresponding author.

[2]1) $D = [0, 1e4]$ means dataset $D$ constains all number from 0 to 10000. 2) $D = sample([0, 13333], 10000)$ means sample a subset $D$ with 10000 of size from $[0, 13333]$ uniformly at random

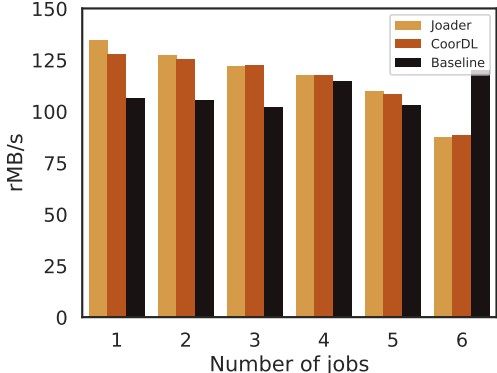

Figure 1: When training multiple ResNet18 models simultaneously, the I/O speed is shown above.

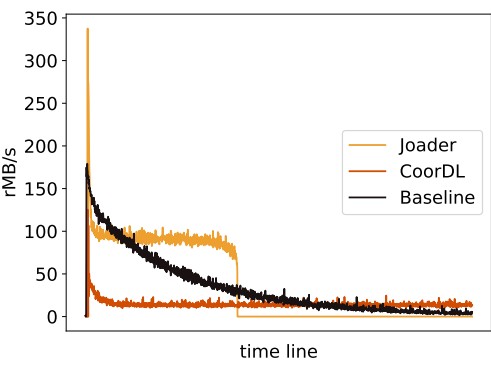

Figure 2: When training 5 models with different speeds at the same time, the I/O speeds are shown above.

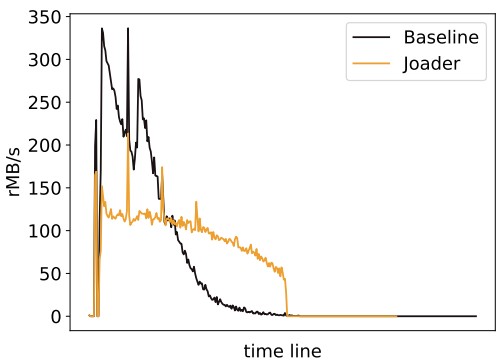

(a) When four jobs start at different moments, the I/O speeds are shown above.

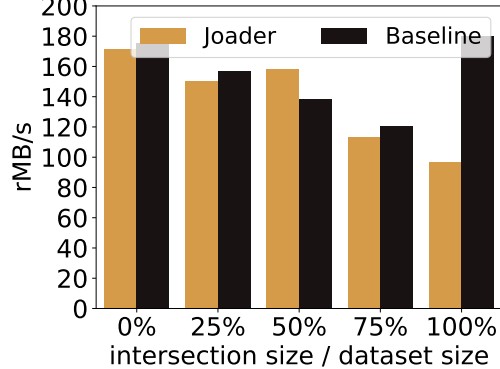

(b) I/O speed are shown above when intersection sizes are different.

Figure 3: Training jobs arrive at different time and have different datasets

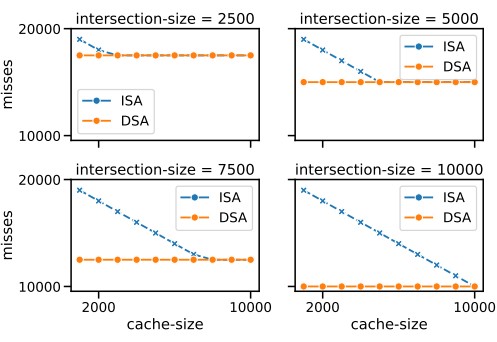

(a) Misses count of DSA and ISA for different intersections with a cache of size from 1 to 10000. DSA can always get the minimum misses.

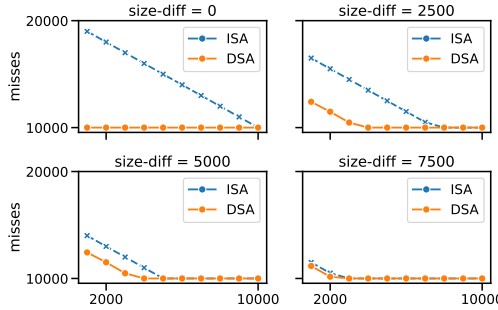

(b) Misses count of DSA and ISA for different dataset sizes with a cache of size from 1 to 10000. DSA can reduce lots of misses.

Figure 4: Misses count of DSA and ISA in various scenes for 2 jobs

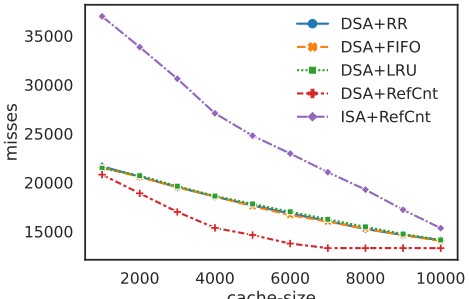 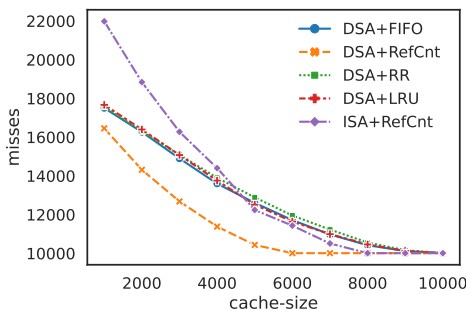

(a) Misses count for 4 jobs in an utterly random scene. RefCnt can reduce 10% misses for the same cache size.

(b) Misses count for 4 jobs with datasets of different sizes. RefCnt can reduce 10% misses for the same cache size.

Figure 5: The number of misses with different cache policies applied with dependent sampling algorithm.

Table 1: Configuration of the number of jobs and datasets

| Algorithm | Case | Num | way of construct dataset |
|---|---|---|---|
| Sampling Algorithm | overlapping partially | 2 | $D_1 = [0, 1e4]; D_2 = [0, 1e4]$ |
| | | 2 | $D_1 = [0, 1e4]; D_2 = [7500, 17500]$ |
| | | 2 | $D_1 = [0, 1e4]; D_2 = [5000, 15000]$ |
| | | 2 | $D_1 = [0, 1e4]; D_2 = [2500, 12500]$ |
| | | 4 | $D_1, D_2, D_3, D_4 = sample([0, 13333], 10000)$ |
| | varying in size | 2 | $D_1 = [0, 1e4]; D_2 = [0, 1e4]$ |
| | | 2 | $D_1 = [0, 1e4]; D_2 = [0, 7500]$ |
| | | 2 | $D_1 = [0, 1e4]; D_2 = [0, 5000]$ |
| | | 2 | $D_1 = [0, 1e4]; D_2 = [0, 2500]$ |
| | | 4 | $D_1 = [0, 1e4]; D_2 = [0, 7500], D_3 = [0, 5000], D_4 = [0, 2500]$ |
| Cache Policy | overlapping partially | 4 | $D_1, D_2, D_3, D_4 = sample([0, 13333], 10000)$ |
| | varying in size | 4 | $D_1 = [0, 1e4]; D_2 = [0, 7500], D_3 = [0, 5000], D_4 = [0, 2500]$ |

sizes and different intersections. We use DSA to access them in shuffling order from the storage with the cache of different sizes (e.g., from one-slot to holding all datasets) to evaluate the number of misses, compared with the experimental results with ISA (shuffling independently).

We start the the evaluation in two jobs training on $dataset_1$ and $dataset_2$, respectively. We set up eight different configurations for the two datasets, which can be divided into two types: the different sizes of intersections and different sizes of the datasets, as shown in Table 1.

For different intersection sizes, Figure 4a plots the relations between cache size and the count of cache misses. For ISA, the misses count is 2x higher than the size of dataset in the case of one-slot cache, shows that all elements are missed and it decreases linearly as the cache size increase. Unless the cache can hold all the data of intersection, the misses count could be minimized. However, DSA can always get the minimum cache misses regardless of the cache size. The reason is that in the selecting procedure, all jobs must select the same subset when the datasets' sizes are equal, due to the conditional probability of 1. Thus, their sampling results would be the same, meaning that all jobs will access the same element simultaneously when the datasets with the same size.

Figure 4b plots the evaluation results on for the datasets with different sizes. For one-slot cache, DSA can reduce 50% cache misses compared with ISA when the dataset sizes differ by 25%. The cache held 30% dataset could get the best performance for DSA while ISA needs to cache 75% dataset to reach optimal. DSA needs to cache 25% data because the size difference brings some uncertainty in the selecting procedure.

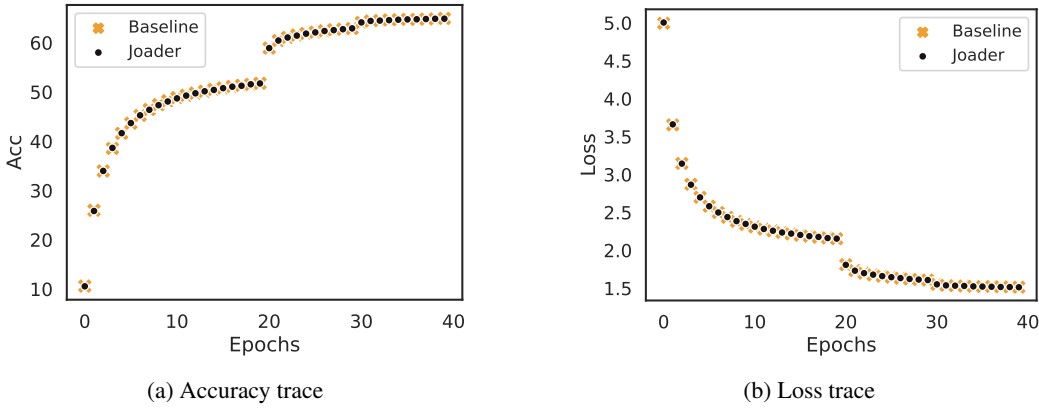

(a) Accuracy trace  (b) Loss trace

Figure 6: Trace of training ResNet18 in 40 epochs

Then, we evaluate the DSA algorithm on four DNN training jobs to show the outstanding performance of the DSA algorithm on multiple datasets. Multiple datasets bring more complexity to the intersection as there can be $2^n - 1$ intersection sets for n sets. To simulate an utterly random scene, we random sample 10,000 elements from 13,333 elements four times, making four different datasets for evaluation. Compared ISA in the random scene, DSA can reduce 50% misses with a one-slot cache (20000 misses out of 40000 misses), as shown in Figure 6a.

DSA also provides good performance for multiple jobs training on datasets with different sizes. Compared to ISA in Figure 6b, one-slot cache can reduce 30% cache misses (16000 misses out of 22000 misses) with DSA when four jobs are training on four datasets, with a 25% different size.

**RefCnt cache policy.** In this part, we compare cache policy RefCnt with the generic cache policies LRU, FIFO, and random replacement (RR) to show 10% misses reduction with the same cache size. We evaluate these policies in above two situations of four jobs: utterly random scene and different dataset sizes. Figure 5 shows the results, which can be summarized as two improvements: 1) RefCnt cache policy can reduce 10% misses in the same cache size compared with general cache policies, and 2) RefCnt cache policy can get the best performance with only caching the 60% dataset while the generic cache policies require holding the whole dataset.

### A.3  Correctness experiments

To demonstrate JOADER does not affect convergence speed w.r.t. accuracy and loss, we train ResNet18 on ImageNet-1K with JOADER and PyTorch[3]. Figure 6 shows that the loss descent trajectory and accuracy trajectory are almost the same.

## B  Dependent Sampling Algorithm

In this section, we show the pseudo-code of the DSA, which involves two functions: selecting and sampling. The selecting algorithm is shown in Algorithm 1, and the sampling algorithm is shown in Algorithm 2.

## C  Sampling in Dependent Sampling Tree

The *sampling* operation involves two steps: 1) *Deciding:* choose a vertex for each job, and 2) *Sampling:* select an element in the corresponding vertex for each job randomly.

In *deciding*, we start the procedure from the root of the tree. In the beginning, all jobs are marked as *undecided*. In each vertex, all undecided jobs need to decide whether to select the current vertex, i.e., the intersection $I$ in the DSA algorithm. If the job chooses the current vertex, we will remove them

---

[3]To activate DSA, we run 2 jobs at different moments

**Algorithm 1:** Selecting Procedure

**Data:** $n$ jobs $J_1, ..., J_n$ and $n$ datasets $D_1, ..., D_n$
**Result:** The subset that each job selects

1   $JOB \leftarrow [J_1, ..., J_n]$; // `Array of all jobs`
2   $DS \leftarrow [D_1, ..., D_n]$; // `Array of all datasets sorted by their cardinality`
3   $S \leftarrow \emptyset$; // `Set of data that are not selected`
4   **while** $JOB \neq \emptyset$ **do**
5      $I \leftarrow \bigcap D_i - S, \forall D_i \in DS$ ;
6      $c \leftarrow |I|$;
7      **for** $D, J$ **in** $DS, JOB$ **do**
        // $\frac{c}{|D|-|S|}$ `is the conditional probability` $\frac{|D_{i-1}|}{|D_i|}$
8         **if** $\mathbf{rand}(0,1) \leq \frac{c}{|D|-|S|}$ **then**
9            $J$ selects $I$;
10           $DS \leftarrow DS - D$;
11           $JOB \leftarrow JOB - J$;
12           $c \leftarrow |D|$;
13         **else**
14           break;
15      **if** *There is no job selecting $I$* **then**
16         $JOB[0]$ selects $D[0] - I$;
17         $DS \leftarrow DS - D[0]$;
18         $JOB \leftarrow JOB - JOB[0]$;
19      $S \leftarrow S \cup I$ ;

---

**Algorithm 2:** Sampling Procedure

**Data:** $n$ jobs $J_1, ..., J_n$ and $k$ subsets $S_1, ..., S_k$
**Result:** The sampling results of each job

1   $JOB \leftarrow [\{J_i, ..., J_j\}, ...]$; // `The job set where job selects the correspond subset`
2   $SUBSET \leftarrow [S_1, ..., S_k]$; // `The subset that the correspond job set selects`
3   **for** $i$ **in** *0..k* **do**
4      $subset \leftarrow SUBSET[i]$;
5      $job\_set \leftarrow JOB[i]$;
6      Sample $e$ uniformly at random in $subset$;
7      **for** $job$ **in** $job\_set$ **do**
8         $job$ picks $e$;

---

from the undecided job set. Then, the remaining undecided jobs are put down to the child and make decisions recursively until the undecided job set is empty, like Algorithm 1.

However, in *deciding*, there may be an intersection that contains multiple vertices of the sampling tree. If job $J_2$ with $D_2$ and job $J_3$ with $D_3$ do not choose the root $X$ in the first recursion, then they should make decision in the intersection $(D_2 - D_X) \cap (D_3 - D_X)$, which contains the datasets in vertices $A$ and $B$. In this case, we should combine these vertices into a big set first and start the deciding procedure once. If someone chooses the big set, it should choose a vertex from the big set randomly again.

After the deciding procedure, each job is assigned to a vertex randomly. Then, we need to select an element in the corresponding vertex randomly. Based on the *united sampling* discussed in Section 3, those jobs that are assigned to the same vertex should share the same sampling result.

Each data item in the dataset should be read exactly once for a job in an epoch. Intuitively, the data item should be deleted from the vertex to avoid being read by the job repeatedly. This process raises the issue that other jobs may need the deleted element that some have read. For example, job $J_1$

Table 2: Notation in sampling algorithm

| | |
|---|---|
| $J_i$ | The i-th job |
| $D_i$ | Dataset that $J_i$ trains upon |
| $I$ | Intersection of all datasets |
| $D_{di}$ | $D_i - I_i$ |
| $|D|$ | Cardinality of set D |
| $J_i^I$ | Event that $J_i$ chooses the intersection |
| $J_i^D$ | Event that $J_i$ do not choose the intersection |
| $e_j^i$ | Element $e_j$ picked by $J_i$ in one round |
| $S$ | Set of elements that are not selected |

fetches the element $e_1$ from vertex $A$ while job $J_2$ and job $J_3$ fetch the element from other vertices. Thus, for $J_1$, we should remove $e_1$ from vertex $A$, but $J_2$ and $J_3$ still need the element $e_1$.

The compensation procedure is introduced in sampling to solve this issue, which adds the deleted element to the child vertex of the job that still needs it. To decide which vertex needs compensation, we construct a job set for each vertex that contains all the jobs that will sample in this vertex. For example, the job set of vertex $A$ is $\{J_1, J_2, J_3\}$ and the set of vertex $B$ is $\{J_2, J_3\}$. And after deleting some elements, we should construct a compensation set containing all the jobs that still need the deleted elements. If the compensation set includes the job set, the deleted elements are added to the corresponding vertex.

**Partially Sampling.** When some jobs are much faster than other jobs, we only need the sampling results for these fast jobs. The sampling procedures are the same as the above. However, the tree cannot be ordered after sampling because these datasets decrease fast and can be less than the above datasets in the tree soon. Therefore, we need to reorder the tree, just like the intersection procedure.

# D  Proof of Algorithm

In this section, we present the derivation and the proof for the proposed algorithm. Proposition 1 shows the first principle for randomness. Table 2 presents some notations for the following proof.

**Proposition 1** *For any job $J_i$, the probability of choosing any element from dataset $D_i$ is $\frac{1}{|D_i|}$.*

## D.1  Derivation in Two-job Case

There are 2 jobs $J_1$ and $J_2$ which training upon datasets $D_1$ and $D_2$, while the intersection is $I = D_1 \cap D_1$ and two difference sets are $D_{d1} = D_1 - I, D_{d2} = D_2 - I$. Their cardinalities are $|D_1|, |D_2|, |I|, |D_{d1}|,$ and $|D_{d2}|$. The event that $J_1$ selects the intersection is $J_1^I$, while the event that $J_1$ selects the difference set is $J_1^D$.

Due to the randomness, the constraints that we need to maintain are

$$\begin{cases} p(J_1^I) = \dfrac{|I|}{|D_1|}, p(J_1^D) = \dfrac{|D_{d1}|}{|D_1|} \\ p(J_2^I) = \dfrac{|I|}{|D_2|}, p(J_2^D) = \dfrac{|D_{d2}|}{|D_2|}. \end{cases} \tag{1}$$

Since $J_2$ makes the decision conditioned on $J_1$, we can the formulate the equations according to the law of total probability in below

$$\begin{cases} p(J_2^I) = p(J_2^I|J_1^I) * p(J_1^I) + p(J_2^I|J_1^D) * p(J_1^D) \\ p(J_2^D) = p(J_2^D|J_1^I) * p(J_1^I) + p(J_2^D|J_1^D) * p(J_1^D). \end{cases} \tag{2}$$

After *shared sampling*, our target is to maximize the probability of $J_1$ and $J_2$ both choose the intersection, which is positive correlated with $p(J_2^I|J_1^I)$. The $p(J_2^I|J_1^I)$ is

$$p(J_2^I|J_1^I) = \frac{p(J_2^I) - p(J_2^I|J_1^D) * p(J_1^D)}{p(J_1^I)}$$

$$\leq min(1, \frac{p(J_2^I)}{p(J_1^I)}) \tag{3}$$

$$\leq min(1, \frac{|D_1|}{|D_2|}).$$

Therefore, we can get the conditional probability in two cases

$$p(J_2^I|J_1^I) = \begin{cases} 1, & \text{if } |D_1| > |D_2| \\ \frac{|D_1|}{|D_2|}, & \text{if } |D_1| \leq |D_2|. \end{cases} \tag{4}$$

When $|D_1| \leq |D_2|$, the other conditional probabilities of $J_2$ conditioned on $J_1$ can be derived as follows

$$\begin{cases} p(J_2^I|J_1^D) = 0 \\ p(J_2^D|J_1^D) = 1 \\ p(J_2^D|J_1^I) = 1 - \frac{|D_1|}{|D_2|}. \end{cases} \tag{5}$$

In *dependent selecting* procedure, the probability of both choosing intersection set is $p(J_1^I)*p(J_2^I|J_1^I)$. With *data partition*, we can get

$$p(e_j^1 = e_i^2) = p(J_1^I) * p(J_2^I|J_1^I) \leq \frac{|I|}{max(|D_2|, |D_1|)}, \tag{6}$$

while $e_j^1, e_i^2$ is the sampling results of $J_1, J_2$ in one round.

We then prove the the proposed dependent algorithm gives the optimal solution for the scenario of two-jobs.

**Proof 1** *Assume that the distribution of the event exists, and the probability of $p(e_j^1 = e_i^2)$ is unknown. When $e_j^1$ and $e_i^2$ are equal, they must belong to the intersection set $D_i$. Therefore, $p(e_j^1 = e_i^2)$ is equal to $p(e_j^1 = e_i^2, e_j^1 \in D_i, e_i^2 \in D_i)$. Due to the definition of joint probability, we can get*

$$p(e_j^1 = e_i^2) \leq p(e_j^1 \in D_i), \quad and \quad p(e_j^1 = e_i^2) \leq p(e_i^2 \in D_i).$$

*Due to the constraint of randomness, $e_j^1$ and $e_i^2$ are both sampled under the uniform distribution, so $p(e_j^1 \in D_i) = \frac{|D_i|}{|D_1|}$ and $p(e_i^2 \in D_i) = \frac{|D_i|}{|D_2|}$. Then $p(e_j^1 = e_i^2) \leq \frac{|D_i|}{|D_1|}$ and $p(e_j^1 = e_i^2) \leq \frac{|D_i|}{|D_2|}$ both hold. Thus, for any distribution of the event, it holds that*

$$p(e_j^1 = e_i^2) \leq \frac{|D_i|}{max(|D_2|, |D_1|)}.$$

### D.2 Derivation in N-job Case

Assume there are $n$ jobs $\{J_1, ..., J_n\}$ and $n$ datasets $\{D_1, ..., D_n\}$, while the intersection $I = \bigcap_{i=1}^{n} D_i$. For the $k$-th job where $k > 1$, the probability of the job $J_k$ chooses the intersection is

$$p(J_k^I) = p(J_k^I|J_1^I, J_2^I, ..., J_{k-1}^I) * p(J_1^I, J_2^I, ..., J_{k-1}^I) + x, \tag{7}$$

where x is the sum of probabilities of $J_k$ chooses intersection in other conditions. With the similar process in Proof 1, we can get the result for $k$-jobs, where the maximum probability of they all choose

intersection is also $p(J_1^I ... J_k^I) \leq \frac{|I|}{max(D_1,...,D_k)}$. Then, we can get

$$p(J_k^I | J_1^I, J_2^I, ..., J_{k-1}^I) \leq min(\frac{p(J_k^I)}{p(J_1^I, J_2^I, ..., J_{k-1}^I)}, 1)$$

$$\leq min(\frac{\frac{|I|}{|D_k|}}{\frac{|I|}{max(D_1,...,D_{k-1})}}, 1) \tag{8}$$

$$\leq min(\frac{max(D_1, ..., D_{k-1})}{|D_k|}, 1).$$

Assume that the datasets are sorted in ascending order w.r.t. their cardinalities, then the formula can be written as for the maximum probability

$$p(J_k^I | J_1^I, J_2^I, ..., J_{k-1}^I) = \frac{|D_{k-1}|}{|D_k|}. \tag{9}$$

The probability of $n$ jobs all choose intersection $I$ is

$$p(J_1^I, J_2^I, ..., J_n^I) = \frac{|I|}{|D_1|} * \frac{|D_1|}{|D_2|} * ... * \frac{|D_{n-1}|}{|D_n|} = \frac{|I|}{|D_n|}, \tag{10}$$

which is the theoretical maximum according to a similar Proof 1.

### D.3 Proof of Randomness

We prove the Proposition 1 for Algorithm 1. Although multiple jobs share the sampling results in the sampling procedure, the random sampling is not changed. Therefore, to prove Proposition 1, we only need to prove the subsets are randomly selected for each job. The Lemma is stated as follow

**Lemma 1** *In each loop of selecting procedure, the intersection $I$ is selected with the probability of $\frac{|I|}{|D_i|}$ for every job $J_i$.*

In the line 8-12 of Algorithm 1, the job $J_i$ chooses the intersection only if the previous job has chosen the intersection $I$, that the probability is

$$p(J_i^I) = p(J_i^I | J_{i-1}^I, ...) * p(J_{i-1}^I, ...)$$

$$= \frac{|I|}{|D_i| - |S|}. \tag{11}$$

Therefore, the Lemma 1 is satisfied only if the probability of entering this loop is $\frac{|D_i| - |S|}{|D_i|}$, that is

**Lemma 2** *For each loop, the set of elements that are not selected is S, and the probability of entering this loop is $\frac{|D_i| - |S|}{|D_i|}$ for job $J_i$.*

**Proof 2** *We prove the Lemma 2 by induction on loop index $k$.*

*Base case. Show Lemma 2 holds for the first loop.*
*The first loop ($k = 1$) must be entered, while $S$ is an empty set and $\frac{|D_i| - |S|}{|D_i|} = 1$. Thus Lemma 2 is satisfied.*

*Induction step. Show that for any $k \geq 1$, if the $k$-th recursion holds Lemma 2, then $(k + 1)$-th loop also holds.*
*The probability of entering the $k$-th loop is $\frac{|D_i| - |S|}{|D_i|}$, and if job $J_i$ does not select the intersection in*

*the k-th loop, then it entering the $(k+1)$-th loop. The probability of $J_i$ does not select intersection is*

$$
\begin{aligned}
p(J_i^D) &= p(J_k^D) + p(J_{k+1}^D) + ... + p(J_i^D) \\
&= p(J_k^D) + p(J_{k+1}^D|J_k^I) * p(J_k^I) + ... + p(J_i^D|J_{i-1}^I) * p(J_{i-1}^I) \\
&= p(J_k^D) + (1 - p(J_{k+1}^I|J_k^I)) * p(J_k^I) + ... + (1 - p(J_i^I|J_{i-1}^I) * p(J_{i-1}^I) \\
&= p(J_k^D) + p(J_k^I) - p(J_{k+1}^I) + p(J_{k+1}^I) - ... - p(J_i^I) \\
&= 1 - p(J_i^I) \\
&= 1 - \frac{|I|}{|D_k| - |S|} * \frac{|D_k| - |S|}{|D_{k+1}| - |S|} * \frac{|D_{i-1}| - |S|}{|D_i| - |S|} \\
&= \frac{|D_i| - |S| - |I|}{|D_i| - |S|}
\end{aligned}
\tag{12}
$$

*where $J_k$ is the first job in job List $JOB$, then we can get the probability of entering the next loop is $\frac{|D_i| - |S| - |I|}{|D_i|}$ while new $S$ is $S \cup I$, which also satisfies the Lemma 2.*

***Conclusion.*** *Since both the base case and the inductive step have been proved as true, the Lemma 2 holds for every loop by mathematical induction.*

## E   System Implementation

In this section, we describe the implementation of JOADER, which is a data loading management system for multiple DNN training jobs. The proposed dependent sampling algorithm applied with the dependent sampling tree is the core in JOADER. In details, we describe the system overview, three main components, and the adaption for distributed DNN training.

### E.1   Overview

**API.** JOADER API allows users to do two things: 1) creating and altering datasets, and 2) registering DNN training jobs. Before training, the user needs to create the dataset with a name in JOADER first. Then, to execute the training job on the dataset, the job should register itself to JOADER with the dataset's name. If multiple jobs are training on that dataset, JOADER will attach the dataset to the jobs for sharing reading and preprocessing. Each job in an epoch will be assigned to a unique id in JOADER, and JOADER will dispatch data according to the job id.

**Architecture.** JOADER includes a frontend and a backend. They communicate with each other in RPC. Figure 7 shows the architecture of JOADER, which consists of three components: Sampler, Loader, and Cache.

### E.2   Sampler and Loader

The Sampler is responsible for data sampling in JOADER. An instance of the Sampler consists of a sampling tree for a dataset. The datasets are organized in tables like the relational database. Each data tuple has a unique integer id (i.e., primary key) to identify itself, consisting of multiple elements, e.g., image, label, and the bounding boxes in ImageNet.

Each job registered to the sampler needs to specify the job's name, dataset's name, names of the needed columns in the table, and a predicate for filtering. Then, the sampler collects the ids of data tuples that meet the requirements and inserts the ids into the sampling tree corresponding to the dataset. Each sampling tree should pick the element uniformly at random in the DS algorithm. These elements (id of data tuple) will be transformed into the data requests according to the columns of each job needed. If the elements and the columns are the same, these data requests will be merged into one request.

The Loader is responsible for reading and processing data w.r.t. the data request from Sampler. In JOADER, each loader corresponds to one type of storage, e.g., POSIX file system, key-value database, or distributed storage. Loaders encapsulate different storages and provide a unified interface to Sampler to present good scalability and compatibility.

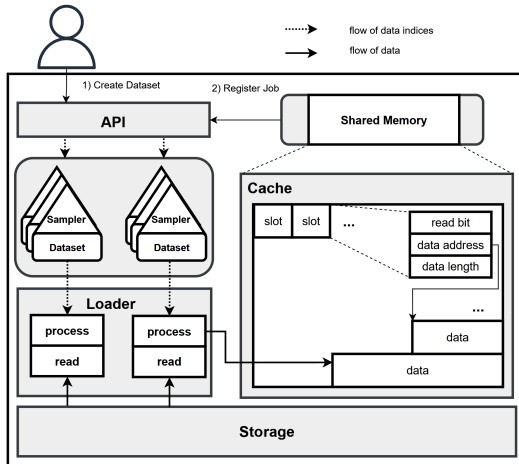

Figure 7: The architecture of JOADER.

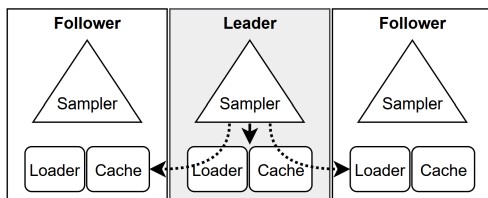

Figure 8: JOADER for distributed DNN training, where the leader JOADER is responsible for the sampling globally.

## E.3 Specific Cache Implementation

The Cache is used to store and share data between JOADER and the DNN training jobs, which is implemented by shared memory to avoid the cost of memory copying, serialization, and network. To manage data and reduce data race, the layout of cache is slotted, making multiple slots in the head of the cache, as shown in Figure 7. Each element in the cache is managed by a slot containing the start address, length, and read bit of the data. The read bit is used to determine whether the data has been consumed.

When Loader tries to load data, it needs to require slots and a contiguous block memory for storing this data. Then, the Loader dispatches the slot ids to training jobs that need the data. These jobs will access the data soon and set the read bit. When there is not enough memory, the reclamation program should be triggered. Data can be classified as consumed and unconsumed in the cache according to the read bit. Because those unconsumed data were sampled earlier than the data that Loader is trying to require memory, they will be accessed earlier than the new data and cannot be evicted.

We can only reclaim those consumed elements, which are prioritized in different levels. For example, there are two training jobs $J_1$ and $J-2$ all want to read four data $\{d_1, d_2, d_3, d_4\}$. In the first round of sampling, $d_1$ is consumed by both jobs. In the second round, job $J_1$ consumed $d_2$ while job $J_2$ consumed $d_3$. Then the memory reclamation program is triggered. Considering the constraint that each job should traverse the dataset once. In this time, data $d_1, d_2, d_3$ are in cache while data $d_1$ is no longer needed anymore while $d_2, d_3$ are still needed. Therefore, $d_1$ will be prioritized for eviction.

In JOADER, we reclaim data according to the number of reference of data.

## E.4 Towards Distributed Training

In model parallelism, the model is segmented into different parts that can run concurrently in different nodes. Only one node is responsible for data preparation. Therefore, JOADER need not change anything to apply to the distributed training in model parallelism.

In data parallelism, the dataset is divided into several partitions, where the number of partitions is equal to the total number of available nodes in the cluster. The model is replicated to the worker nodes. Each worker operates on its subset of the dataset to train the model locally. For each worker node, we need to set up a JOADER for sharing data preparation to multiple DNN training jobs in this node. However, the sampling work should be done globally to avoid redundant sampling.

To solve the above issue, we set multiple JOADER for multiple nodes, while only one leader JOADER can sample. The leader needs to dispatch the sampling results to the followers, as shown in Figure 8. During training, each sub-process of the distributed training job in data parallelism needs to register itself to the local JOADER in the same worker node with the same name and the assigned id. Meanwhile, the local JOADER should register a sub-sampler in leader JOADER. The local JOADER should keep fetching sampling results from the leader JOADER and load them to the local cache. By doing this, only leader JOADER is responsible for sampling and dispatching sampling results to followers. Notice that the cache is also distributed in different worker nodes, storing more data than a single cache.

The same data is sent to the same worker node when the leader dispatches sampling results. PyTorch uses the hash partition algorithm to dispatch data. For example, suppose a worker node array $nodes = [node_1, node_2, ..., node_n]$. The data with integer $id$ is sent to $nodes[id\%n]$. However, things will be more complicated when dealing with multiple training jobs. For example, two training jobs are on the worker node arrays $[node_1, node_2, ..., node_{n-1}]$ and $[node_2, node_3, ..., node_n]$, respectively. Due to the offset between the two arrays, the same data cannot be sent to the same worker node for the two jobs by one hash partition.

In JOADER, we solve it by hashing the data id twice. In the first hash, we try to dispatch the data to all the worker node in cluster. Therefore, the same data can be sent to the same node. However, some training jobs are not training on some worker nodes and the data can not be sent to these nodes for these jobs. Therefore, in the second hash, for the data dispatched wrongly, we will try to dispatch it to the nodes locally w.r.t. the configuration of each job.