# OpenReview forum: "A Deep Learning Dataloader with Shared Data Preparation"
_NeurIPS.cc/2022/Conference — NeurIPS 2022 Accept_

### Official Review · Reviewer_Hv96 · 2022-07-04

**Rating:** 5
**Confidence:** 3
**Soundness:** 3 good
**Presentation:** 3 good
**Contribution:** 3 good

**Summary:**

This paper presents Joader, a data loading system optimized for the scenario where multiple training jobs share overlapped sources of data and data preprocessing. The proposed system overcomes the constraints where training jobs could vary in training speed which causes cache thrashing.


**Questions:**

Is there any specific reason that Joader only uses ResNet series in experiments? While image classification is a quite standard task to test against, it would be more convincing to justify with data that it works with diverse set of models.


**Ethics Review Area:**

["I don’t know"]

**Limitations:**

While the proposed system is general and less restrictive comparing with previous work, the experiments are set to specifically justify the claims in previous sections, the reviewer believes that it would potentially attract more users if there is at least one experiment for Joader to focus on distributed training setting.


**Strengths And Weaknesses:**

Strength
1) The proposed idea, dependent sampling algorithm (DSA), is novel and effective with good theoretical guarantee, while providing flexibility in varying speed across training jobs, free starting/stopping of jobs, partitial overlapping;
2) The engineering contribution that integrates Joader into PyTorch and enables distributed training is highly convenient for downstream users of this research;
3) Preliminary experiments demonstrate that this approach is effective.

Weaknesses:
1) The evaluation is only conducted on a single workload or a single series of workloads (ResNet).
2) Distributed training does not exist in experiment setting, while it is indeed an extremely important usecase considering multiple overlapped training jobs in a cluster. Appendix D.4 does briefly mention that there is no much change to make Joader work for distributed training, but no further experiments are conducted.

---

> ### Author Response · Authors · 2022-08-02
> **Response to Reviewer Hv96**
>
> **Weakness_1&Question:**
>
> Essentially, we choose the ResNet family although our method is independent of the model structure, and our proposed method is only related to the model training speed (the training speed varies depending on the number of ResNet layers). To further demonstrate the versatility of our method, we have conducted additional evaluations on the other sets of models. For each model, we run six workloads in parallel and compare Joader with the baselines. The results show that our Joader consistently outperforms the baseline when multiple jobs are running in parallel (see below).
>
> |model|time in baseline(min)|time in Joader(min)|
> |:-|-:|-:|
> |**AlexNet**|72.68671|40.01698|
> |**SqueezeNet**|71.99047|40.59335|
> |**ShuffleNet**|82.02182|47.65184|
> |**MobileNet_v2**|75.24602|42.39433|
> |**MobileNet_v3_small**|63.79987|38.60851|
>
> **Weakness_2:**
>
> Joader for distributed training is another important use case that requires an independent future study.
> In this paper, our goal is to propose the sampling algorithm with a specific data structure, which is independent of the system implementation. In this regard, evaluation on a single machine is sufficient to verify the performance of our algorithm. The implementation and evaluation of distributed systems add extra complications. It requires additional design on data assignment and communication in a distributed manner, which are not the focus of the sampling algorithm. So we did not implement and evaluate Joader in a distributed system when we submitted this paper. However, we could provide some discussions about the guidance of implementation when training models in terms of data parallelism. In a distributed environment, each node runs a Joader, and we select one Joader as the leader which is responsible for sampling globally in DSA. The leader is in charge of dispatching the data indices to each follower Joader, while these followers can maintain the cache and load the data from their local disks.

---

### Official Review · Reviewer_6QhD · 2022-07-11

**Rating:** 8
**Confidence:** 5
**Soundness:** 4 excellent
**Presentation:** 3 good
**Contribution:** 4 excellent

**Summary:**

The authors revealed an interesting and practical problem in training parallel DNNs. The redundant consumption of I/O and data preparation, which seems not as crucial as computation in GPU, affects the DNN training speed, especially when multiple DNNs are training parallelly in a GPU server. To solve this problem, the authors proposed a new data loading method for efficiently training parallel DNNs. The method includes a data sampling algorithm to increase the sampling locality with guaranteed randomness. For practical usage, a data structure-dependent sampling tree and a specific cache. Altogether, a prototype Joader (integrated with PyTorch) is implemented to show the extremely fast performance in training DNNs parallelly.

**Questions:**

1.	 If I understand clearly, the proposed method totally controls to sample the same data when the datasets are the same. Compared to the classic sampling in random, is there a tradeoff between strict correlation and randomness?
2.	Is it possible to give the proof to the optimality in N-job case?


**Limitations:**

N.A.

**Strengths And Weaknesses:**

Strengths:
1.	The problem is practical and essential to society. The novelty of the method is clear enough. The method keeps the characteristic of random sampling for each DNN training job, and reduces the redundant consumption of I/O and data preparation.
2.	The theoretical proof guarantees the correctness of the algorithm, and the algorithm is proved to reach global optimal in the two-job case.
3.	The authors implemented a prototype Joader, which is integrated with PyTorch. The details of the system design and implementation are provided in Appendix.
4.	The evaluation results clearly show the superiority of training speed improvement from implemented Joader.


Weaknesses
1.	The description of the N-job case is a little bit complicated. After carefully reading the operations of the dependent sampling tree and the pseudo-code of DSA, I could finally understand the N-job case.
2.	The authors proved the optimality in the two-job case, which is very important for this problem theoretically. The algorithm in the N-job case seems to be a greedy algorithm. It is better that the authors could give proof of the optimality.

---

> ### Author Response · Authors · 2022-08-02
> **Response to Reviewer 6QhD**
>
> **Weakness_1:**
>
> For the description of the N-job case, we will revise it to describe the algorithm more clearly and concisely.
>
> **Weakness_2 & Question_2:**
>
> For the optimality in the N-job case, we can indeed get the optimum at each step  (a loop in pseudo-code, i.e., line 5 to line 14  or a level in Figure3).
>
> Proof:
>
> Assume there are n jobs $\{J_1,...,J_n\}$ training upon n datasets $\{D_1,…,D_n\}$.
>
> In each step, we have proved that for $k$-th job in each loop, the probability  of choosing intersection $I$ is
>
>
> $$
> p =\frac{I}{D_k}
> $$
>
> in Lemma1 of Appendix C.3. The probability is obviously optimal because it is equal to the probability that $J_k$ randomly chooses intersection $I$ directly.
>
> However, the global optimal probability is associated with the number of jobs, the sizes of the intersections,  the sizes of the datasets, and the shape of the sampling tree (left-deep tree or bushy tree). Considering the algorithm is a greedy strategy, it is hard to prove the global optimality for now. Nevertheless, we believe the proposed method with the optimality for the 2-job case and the superior efficacy in practice could contribute to the community and is worthy of further exploration.
>
> **Question_1:** For the tradeoff between strict correlation and randomness, the goal of this paper is to efficiently load the data while retaining randomness for each job. So the proposed method will get a strict correlation for the jobs with the same datasets. The balance between strict correlation and randomness among jobs could be a good future direction. On our conjecture, if the randomness could be loosened to some degree (which could be measured), the sampling performance could be further improved with a guarantee in terms of the metric of the degree.

---

> > ### Comment · Reviewer_6QhD · 2022-08-09
> > **Further comments after rebuttal**
> >
> > I totally understand it is hard to prove the optimality in the N-job case in such a practical scenario. The evaluation in the paper and your comments convinced me that the contributions of the DSA's current version are enough for the community. Measuring the degree of breaking the strict correlation among jobs may also be a good direction to study in the future.
> >
> > I also read the replies about the distributed scenarios to the other reviewers. This paper reveals an important problem, and proposes an algorithm with the specific data structure and Cache policy to deal with the problem. The theoretical proofs for the algorithm, and the evaluation for the implemented Joader show the superiority. Regarding the algorithm, I think this paper almost resolves the problem in a classic scenario (i.e., on a server). The implemented Joader is more like a prototype showing the algorithm's performance. I agree that the distributed scenario is important and deserves further study, but implementing the algorithm for a distributed system is another story. It may not be a weakness of this paper.
> >
> > Thus, I would like to champion this paper.

---

### Official Review · Reviewer_GZ15 · 2022-07-12

**Rating:** 5
**Confidence:** 5
**Soundness:** 2 fair
**Presentation:** 3 good
**Contribution:** 3 good

**Summary:**

To efficiently perform multi-task training jobs on overlapped datasets, the authors propose a new data loading method, focusing on data preparation. They design a dependent sampling algorithm(DSA) and domain-specific cache policy to maximize the locality. Moreover, a novel tree data structure is constructed to efficiently implement DSA. The experiments show a greater versatility and superiority of training speed improvement without affecting accuracy.


**Questions:**

What technologies do they use to make CoorDL achieve better performance than Joader in synchronous cases?

**Ethics Review Area:**

["I don’t know"]

**Limitations:**

See above.

Overall, the being solved problem is a significant problem. The approach used in this work looks new. However, the experimentation should be improved. RefCnt Cache policy looks trivial.

**Strengths And Weaknesses:**

Strength:
* Propose a new data loading method for training multiple parallel jobs on overlapped datasets
* Propose a dependent sampling algorithm (DSA) to maximize the sampling locality while ensuring correlated randomness
* Design a tree-based data structure for efficient cache policy implementation
* The experiment results are convincing. Joader reduces end2end multitask training time while keeping CPU utilization low.

Weakness:
* To maximize the sampling locality, the authors need extra operations, including dividing the datasets into subsets and calculating the probabilities and conditional probabilities. With the number of training jobs increasing, the overhead could be a problem and they didn't mention it.
* The motivation is weak. It's a rare case that researchers will train multiple tasks on overlapped datasets on the same machine.

---

> ### Author Response · Authors · 2022-08-02
> **Response to Reviewer GZ15**
>
> **Weakness_1:**
>
> The reviewer asked about the extra overhead of our method, especially when the number of training jobs increases. In Joader, the time complexity of dataset operations, i.e., sampling, deletion, and insertion, are O(1), O(1), and O(|D|), respectively, for each dataset. Since the dependent sampling tree manages the **index** of each input rather than the input itself, executing the operations is extremely fast. For implementation, we use a bitmap to represent the dataset in our dependent sampling tree, which is efficient for these operations. In practice, the operations only count up to a minor fraction of the time consumption of the total data preparation process.
>
> To evaluate the time cost of dataset operations, we have conducted an additional experiment by randomly inserting 128 datasets into our system. The size of each dataset is between 1,000,000 and 2,000,000 elements (note that ImageNet contains 1,400,000 images). The average cost of inserting each dataset is 0.57 seconds. And the time cost of the operations remains nearly constant as the number of datasets increases.
>
> Deletion and sampling are also efficient in our algorithm. Sampling takes an average of  0.000054 seconds for each element, and deletion takes an average of 0.00000105 seconds. We will add the corresponding description and experiment in our rebuttal revision.
>
> **Weakness_2:**
>
> For our motivation, we claim that the scenarios of training multiple tasks on overlapped datasets on the same machine are actually common. We briefly clarify our motivation as follows.
>
> Training multiple models in parallel are typical and practical in HyperParameter Search (HPS) [1,3] and Neural Architecture Search (NAS) [2,4]. Models with different architectures or HyperParameters may lead to different training speeds. No matter the traversal orders on the same dataset, different training speeds make the remaining subsets in an epoch overlap at every moment. Meanwhile, many existing HPS and NAS work runs on a single server with multi-GPUs [5]. The above scenarios motivate us to propose the algorithm for efficient data preparation and accelerating research and AI application development.
>
> **Question_1:**
>
> Sorry for the unclear description of the synchronous case. We clarify the term “synchronous case” in our evaluation. That is, all jobs of Joader and CoorDL[1] train the same model on the same dataset, load data in the same order and start at the same time. We try to make fair comparison by forcing the jobs training on the GPUs of the same version to get the theoretically same training speeds.
>
> CoorDL used a straightforward yet effective strategy. The reason for its performance can be found in the CoorDL paper (Section 6.3, [1]), quoted below:
>
> “Each job in the HP search operates on the same data; hence, instead of accessing data independently for each job, they can be coordinated to fetch and prep the dataset exactly once per epoch. Each epoch is completed in a synchronized fashion by all HP jobs; as a result, preprocessed mini-batches created by one job can be reused by all concurrent jobs.”
>
> We implemented the above method (CoorDL) and compared it with our DSA. However, the DSA is naturally asynchronous. Thus training speeds of the jobs would be inevitably affected by the minor performance distinction among GPUs and the concurrency control of the operating system. Notice that CoorDL forces the jobs’ executions at the same pace. If we add such synchronous control to DSA,  it will perform exactly the same as CoorDL.
>
> **RefCnt policy**:
>
> RefCnt is effective and important. In our revision, the results in Figure 12 show that RefCnt significantly improves performance. Compared to the classical cache policy, the RefCnt policy can reduce up to 50% of cache misses when the cache can hold half of the dataset.
>
> [1] Mohan, Jayashree, Amar Phanishayee, Ashish Raniwala, and Vijay Chidambaram. "Analyzing and Mitigating Data Stalls in DNN Training.” *Proc. VLDB Endow.* 14, no. 5 (2021): 771–84.
>
> [2]Elsken, Thomas, Jan Hendrik Metzen, and Frank Hutter. "Neural architecture search: A survey." *The Journal of Machine Learning Research* 20, no. 1 (2019): 1997-2017.
>
> [3]Feurer, Matthias, and Frank Hutter. "Hyperparameter optimization." In *Automated machine learning*, pp. 3-33. Springer, Cham, 2019.
>
> [4]Liu, Chenxi, Barret Zoph, Maxim Neumann, Jonathon Shlens, Wei Hua, Li-Jia Li, Li Fei-Fei, Alan Yuille, Jonathan Huang, and Kevin Murphy. "Progressive neural architecture search." In *Proceedings of the European conference on computer vision (ECCV)*, pp. 19-34. 2018.
>
> [5]Ben-Nun, Tal, and Torsten Hoefler. "Demystifying parallel and distributed deep learning: An in-depth concurrency analysis." *ACM Computing Surveys (CSUR)* 52, no. 4 (2019): 1-43.

---

### Author Response · Authors · 2022-08-08
**Author-reviewer discussion follow-up**

Dear reviewers,

We thank all the reviewers' detailed and constructive comments and concerns. Since only a few days are left in the author-reviewer discussion, we hope our responses address the reviewers' concerns adequately. In case we miss anything in our responses, please feel free to leave any further comments, concerns, or suggestions to us. We are very happy to discuss and answer further questions here.

Best

---

### Meta-Review · Area_Chair_aupG · 2022-08-23

**Recommendation:** Accept
**Confidence:** Certain

**Metareview:**

The paper proposes a new data loader called Joader for parallel DNN training on overlapped datasets that allows tasks to share memory and computational resources for data preprocessing. Joader implements a new sampling mechanism and cache policy to reduce cache misses due to data access from multiple tasks and a new data structure to facilitate the implementation. Joader has been integrated with PyTorch and shown to be very effective in practice.

All reviewers agree that the paper makes a valuable contribution to the NeurIPS community and I agree with Reviewer 6QhD that the contribution stands even if the system is not covering distributed workloads. I thus recommend acceptance.

However, for the camera ready version I think it is important that the authors incorporate additional discussion about potential limitations of their approach so that the tool can be used most effectively by researchers. For example, potential overheads in the implementation of the new data structure should be discussed even if they are small in the provided experiments and also the fact that sampling across jobs is now correlated and no longer independent is important to emphasize together with potential implications for the resulting workloads (e.g. can the results of the parallel runs still be used to achieve variance reduction by ensembles?). Beyond that I think the additional experiments to break down the performance gains into individual components is valuable and clarification made in the conversation with the reviewers should be incorporated in the paper.

**Award:**

No

---

### Decision · Program_Chairs · 2022-09-14

Accept